# Performance of the Abbott Architect Immuno-Chemiluminometric NT-proBNP Assay

**DOI:** 10.3390/diagnostics12051172

**Published:** 2022-05-08

**Authors:** Chin-Shern Lau, Ya Li Liang, Soon Kieng Phua, Gillian Murtagh, Imo E. Hoefer, Ron H. Stokwielder, Milica Kosevich, Jennifer Yen, Jaganathan Sickan, Christos Varounis, Tar-Choon Aw

**Affiliations:** 1Department of Laboratory Medicine, Changi General Hospital, 2 SIMEI STREET 3, Singapore 529889, Singapore; michael.lau@mohh.com.sg (C.-S.L.); yali_liang@cgh.com.sg (Y.L.L.); soon_kieng_phua@cgh.com.sg (S.K.P.); 2Abbott Laboratories, Abbott Park, IL 60064, USA; gillian.murtagh@abbott.com (G.M.); milica.kosevich@abbott.com (M.K.); jennifer.l.yen@abbott.com (J.Y.); jaganathan.sickan@abbott.com (J.S.); christos.varounis@abbott.com (C.V.); 3Central Diagnostic Laboratory, University Medical Center, 3584 CX Utrecht, The Netherlands; i.hoefer@umcutrecht.nl (I.E.H.); r.stokwielder@umcutrecht.nl (R.H.S.); 4Department of Medicine, National University of Singapore, Singapore 119077, Singapore; 5Academic Pathology Program, Duke-NUS Medical School, Singapore 169857, Singapore

**Keywords:** NT-proBNP, heart failure, immunoassay

## Abstract

**Background:** We evaluated the performance of the Abbott N-terminal pro-brain natriuretic peptide (NT-proBNP) assay against the Roche NT-proBNP immunoassay across two sites. **Methods:** Precision, linearity, and sensitivity studies were performed. A combined method of comparison and regression analysis was performed between the Roche and Abbott assays using samples from both sites (n = 494). To verify biotin interference, lyophilised biotin powder was reconstituted and spiked into serum samples at two medical decision levels (final concentration 500/4250 ng/mL) and compared to controls. NT-proBNP was also measured in anonymised leftover sera (n = 388) in a cardio-renal healthy population and stratified into three age bands—<50 (n = 145), 50–75 (n = 183) and >75 (n = 60). **Results:** Between-run precision (CV%) for NT-proBNP was 4.17/4.50 (139.5/142.0 pg/mL), 3.83/2.17 (521.6/506.3), and 4.60/2.51 (5053/4973), respectively. The assay was linear from 0.7–41,501 pg/mL. The limit of blank/quantitation was 1.2/7.9 pg/mL. The assay showed no interference from biotin up to 4250 ng/mL. Passing–Bablok regression analysis showed excellent agreement between the two assays (r = 0.999, 95% CI 0.999 to 0.999, *p* < 0.0001). The Roche assay had a slightly persistent, negative bias across different levels of NT-proBNP. ESC age cut-offs for diagnosing acute heart failure are applicable for the Abbott assay, with the median NT-proBNP of subjects < 50 years old at 43.0 pg/mL (range 4.9–456 pg/mL), 50–75 years old at 95.1 pg/mL (range 10.5–1079 pg/mL), and >75 years old at 173.1 pg/mL (range 23.2–1948 pg/mL). **Conclusions:** The Abbott Architect NT-proBNP assay has good performance that agrees with the manufacturer’s specifications. ESC/AHA recommended NT-proBNP age groups for acute heart failure diagnosis are applicable to this assay.

## 1. Introduction

N-terminal pro-brain natriuretic peptide (NT-proBNP) is an important cardiac biomarker for heart failure (HF). Physiologically, cardiac stress initiates the synthesis of natriuretic peptide precursors in the ventricular and atrial myocardium [1]. Under these conditions, pre-proBNP (134 amino acids) is converted to the precursor peptide pro-BNP (1–108), which is then cleaved into a bioactive hormone, B-type natriuretic peptide (BNP) (biologically active 32 amino acid C-terminal peptide) and a biologically inactive N-terminal fragment, NT-proBNP (1–76) [2]. Although some other physiological factors may cause discrepant NT-proBNP measurements—for example, a consistent inverse relationship between obesity (BMI ≥ 30) and circulating natriuretic peptide levels has been demonstrated [3]—NT-proBNP remains a core biomarker that serves multiple functions in the management of HF. The European Society of Cardiology recommends NT-proBNP as an initial diagnostic test in all patients presenting with symptoms suggestive of new onset or worsening HF, with a cut-off of <125 pg/mL being used to rule out HF in patients < 50 years old with mild symptoms in the non-acute setting and <300 pg/mL in the acute setting with shortness of breath [4,5]. NT-proBNP has also been established as an excellent marker for the monitoring of therapeutic intervention to guide HF therapy [6]. It may even be used to guide therapy in patients without known cardiovascular disease, as when a cut-off of NT-proBNP > 125 pg/mL was used in high-risk patients with type 2 diabetes mellitus without a known history of cardiovascular disease to decide to intensify anti-hypertensive treatments [7], it resulted in a reduction in the rate of hospitalisation/death due to cardiac disease by 65%. Indeed, its significance has only increased in this regard, with the advent of new HF treatments. Angiotensin receptor neprilysin inhibitors (ARNIs) are a new class of drug that can block both the renin–angiotensin–aldosterone system as well as the breakdown of natriuretic peptides by the enzyme neprilysin. ARNIs have been shown to be superior to enalapril in reducing the risks of death and hospitalisation in HF [8]. NT-proBNP has an advantage over traditional BNP when monitoring patients on ARNIs, as shown in the PARADIGM-HF trial [9]. In this trial, early after the initiation of Entresto (sacubitril/valsartan), BNP paradoxically increased from 202 ng/L to 235 ng/L during the first 8–10 weeks of treatment, whereas NT-proBNP demonstrated an appropriate decline (median reduction in NT-proBNP of 28%) as patients improved. As these new HF medications become more widely used, the use of NT-proBNP will be preferred over BNP. NT-proBNP is also used in the prognostication of both reduced and preserved ejection fraction HF [10], as well as in pulmonary arterial hypertension [11,12]. NT-proBNP shows superior prognostic performance even when compared to risk stratification models [13], being more significantly associated with cardiovascular disease death/all-cause death than the ESC/EASD risk model for both outcomes (C-index: CVD death 0.8 vs. 0.53, all-cause death 0.73 vs. 0.52), with NT-proBNP > 125 pg/mL having a 7.2/3.1-fold risk for cardiovascular disease/all-cause death at 10 years. Even in high-risk patients with diabetes, NT-proBNP by itself was as discriminatory as the model of 20 traditional clinical and laboratory variables in the prediction of both death and cardiovascular events [14]. NT-proBNP has even been demonstrated as an independent predictor of the occurrence and recurrence of atrial fibrillation in patients with silent atrial fibrillation [15], with a receiver operating characteristic area under the curve of 0.76. NT-proBNP can also be used as a surrogate for clinical outcomes in HF trials [16], as well as a useful marker to determine the inclusion of trial patients with probable acute HF in the emergency setting [17].

Furthermore, from the analytical standpoint, the assessment of NT-proBNP has several advantages over BNP. As an analyte, NT-proBNP has a longer half-life than BNP (70 min vs. 22 min) [18] and has longer stability in storage, with studies showing only a <10% decrease in >90% of samples stored at −20 degrees Celsius after two years [19]. As most NT-proBNP assays are based on the same antibodies and calibrators as the Roche assay, the results are more comparable. Although there remains some between method variability, it is much less for NTproBNP compared to the BNP assays. In one Italian proficiency testing program [20], the between-method variability for NT-proBNP was 8.7 CV%, whereas the BNP variability was much higher at 43.0 CV%. BNP can only be analysed on EDTA plasma specimens [21]; however, NT-proBNP can be assessed using serum or plasma, with minimal bias between the two sample types [22], granting some flexibility in terms of sample acquisition.

We have evaluated the performance of the new Abbott NT-proBNP assay on the Architect i2000 analyser (Abbott Laboratories, Illinois, USA) against the Roche electro-chemiluminescence NT-proBNP immunoassay on the Cobas e801 (Roche Diagnostics, Singapore) and Cobas e411 (Roche Diagnostics, The Netherlands) analysers across two sites. In addition, there are concerns that, as a sandwich (non-competitive) immunoassay utilizing biotinylated components, NT-proBNP assays can be susceptible to biotin interference [23]. As such, the Netherlands site performed an assessment of the levels of biotin interference on the Abbott NT-proBNP assay. Furthermore, age-related HF diagnostic thresholds for NT-proBNP are recommended (<50 y: 450 pg/mL, 50–75 y: 900 pg/mL and >75 y: 1800 pg/mL) [24]. A prior study [25] has demonstrated that the age-adjusted Roche NT-proBNP cut-offs showed an even greater discriminative power for acute decompensated heart failure in Singapore when compared to New Zealand, with a higher area under the curve on ROC analysis (0.93 vs. 0.87). However, we are unaware of any age-related cut-offs for the Abbott NT-proBNP assay. Thus, we attempted to determine the applicability of the ICON study age group (<50 y, 50–75 y and >75 y) NTproBNP cut-offs for the rule-in of acute heart failure to the Abbott assay at the Singapore site.

## 2. Methods

### 2.1. Participants

To determine age-appropriate median NT-proBNP levels in Singapore, NT-proBNP was measured in a separate group of leftover sera from 388 (M = 202) de-identified, anonymised samples from subjects without history of heart disease and with eGFR > 90 mL/min. These were sorted into 3 age categories (<50, n = 145; 50–75, n = 183; >75, n = 60) to compare median NT-proBNP levels between age groups in a cardio-renal healthy population.

### 2.2. Materials and Methods

All serum samples used were anonymised and from deidentified leftover sera stored at −70 degrees Celsius, if not immediately analysed. Frozen samples were thawed for 1 h at room temperature just prior to analysis. Thawed samples were vortexed before analysis. Precision, linearity, and sensitivity studies were performed using a procedure based on the Clinical and Laboratory Standards Institute (CLSI) guidelines EP05-A3 [26], EP06-A [27], and EP17-A2 [28]. Within-run imprecision was assessed on 3 levels of Abbott controls (20 results each) at both sites. For between-run precision analysis, a single Architect run was performed over 5 days, with 3 controls tested in 5 replicates every run. For linearity analysis, samples with known high analyte concentrations were selected to a produce levels over a clinically relevant range. For limit of blank determination, 2 blank samples were run 10 times each. For limit of quantitation assessment, 5 low-level NT-ProBNP human samples with concentrations ranging from 1.98 to 15.9 pg/mL were tested over 5 days to generate 20 results each. A combined method comparison and regression analysis (n = 494) was performed from 297 samples (Singapore) and 197 samples (the Netherlands) that were analysed on both the Architect and Roche NT-proBNP assays (n = 494). Auto-dilution verification was performed for the Netherlands site with a 1:2 auto-dilution vs. a 1:2 manual dilution method, using 7 human serum samples spiked with NT-proBNP to produce levels within 75–125% of the upper analytical measuring range. Five replicates each of the auto-diluted and manually diluted samples were assayed on one Architect analyser. The Abbott Architect NT-proBNP immunoassay is a two-step quantitative chemiluminescent microparticle immunoassay, where sample and biotinylated anti-NT-proBNP coated paramagnetic microparticles are combined. The NT-proBNP present complexes with the microparticles, and after wash steps, trigger solutions result in a chemiluminescent reaction, measured as relative light units, that is directly proportional to the NT-proBNP in the sample. The assay has a claimed precision of 3.6% at 150 pg/mL and 5.1% at 29,275 pg/mL, a claimed linear range of the limit of quantitation (8.2 pg/mL) to 35,000 pg/L, limit of blank of 2.6 pg/mL, and a limit of detection of 4.9 pg/mL, with a high dose hook effect occurring at 461,324 pg/mL. The Abbott assay includes biotinylated components, and the manufacturer claims a biotin interference threshold of 4250 ng/mL. To verify the biotin interference threshold of the Abbott assay, the Netherlands site used lyophilised biotin powder (#4401, Sigma-Aldrich, St. Louis, MO, USA) that was reconstituted in DMSO (#D2650, Sigma-Aldrich, St. Louis, MO, USA) and spiked into serum samples at two Medical Decision Levels (MDLs: 80 pg/mL and 1800 pg/mL) at a final concentration of 500 to 4250 ng/mL. Control samples were prepared with an equivalent volume of DMSO. Samples (with and without biotin), calibrators, and controls were tested in replicates of five on the Abbott ARCHITECT i2000SR immunoassay analyser using one reagent and control lot and two lots of calibrators. Controls passed specification in all testing.

The Roche proBNP II assay is a sandwich immunoassay, involving a biotinylated monoclonal NT-proBNP-specific antibody and a monoclonal NT-proBNP-specific antibody labelled with a ruthenium complex and streptavidin-coated microparticles to form a sandwich complex, which is then bound to the solid phase. The reaction mixture is aspirated into the measuring cell where microparticles are magnetically captured onto the surface of the electrode, where a voltage induces a chemiluminescent reaction. It has a reported range of 5–35,000 pg/mL or up to 70,000 pg/mL for 2-fold dilution. The limit of detection of the assay is 5 pg/mL, with a precision of 3.5% at 59.3 pg/mL, and 2.0% at 6552 pg/mL, and a functional sensitivity of CV 20% at 50 pg/mL. Biotin tolerance was reported to be up to 1250 pg/mL since 2019.

### 2.3. Statistical Analysis

Data were presented in either mean ± standard deviation or median (inter-quartile range) where appropriate. No indeterminate or missing results were used. Passing–Bablok regression analysis was also performed to assess the agreement between Abbott and Roche NT-proBNP. Bias was evaluated using the Bland–Altman method. Mann–Whitney U analysis was also performed between age groups in cardio-renal healthy subjects. We used MedCalc Statistical Software (version 20.008, MedCalc Software Ltd., Ostend, Belgium) for statistical analyses. For limit of quantitation analysis, we used a non-linear regression model using GraphPad Prism, version 9.2.0, GraphPad Software, San Diego, CA, USA. As this was part of routine clinical laboratory method evaluation, national regulations exempts such investigations from IRB review. The study was conducted in compliance with STARD guidelines (see Appendix A).

## 3. Results

### 3.1. Performance Evaluation

At both sites, intra-assay precision of the Abbott NT-proBNP was tested at three levels, and within-run precision was 2.84%/4.30% (137.6/142.0), 3.16%/2.17% (505.7/506.3), and 3.61%/2.50% (4980/4973). Between-run precision was 4.17%/4.50% (139.5/142.0), 3.83%/2.17% (521.6/506.3), and 4.60%/2.51% (5053/4973) (see Appendix A). The assay was shown to be linear from 0.7 to 41,501 pg/mL (see Appendix A) and the limit of blank was 1.2 pg/mL (claimed 2.6 pg/mL), with the limit of quantitation being 7.9 pg/mL (claimed 8.2 pg/mL) (see Appendix A). The mean percent recovery of all samples used in auto-dilution verification at the Netherlands site was 99.7%.

### 3.2. Combined Regression Analysis and Method Comparison

Passing–Bablok regression analysis of all 494 samples showed good agreement between the Roche and Abbott NT-proBNP assays (r = 0.999, 95% CI 0.999 to 0.999, *p* < 0.0001) (see Figure 1).

We performed a Bland–Altman analysis of the combined samples at various levels of Roche Cobas NTproBNP (see Figure 2). The Roche had a persistent negative bias compared to the Abbott NT-proBNP, which ranged from 25.5% (15.5 pg/mL) at <125 pg/mL to 12.4% (916 pg/mL) at >1800 pg/mL (see Table 1).

### 3.3. Assay Robustness to Biotin Interference

Analysis of serum samples in the presence and absence of biotin revealed a unit change of ≤0.86 pg/mL and a percentage change of ≤1.18% across the range of biotin concentrations tested for the low MDL, and a unit change of ≤54.68 pg/mL and % change of ≤3.07% across the range of biotin concentrations tested for the high MDL. Table 2 shows the concentration and % difference for each sample. The percentage bias between a control sample and a sample spiked with levels as high as 4250 ng/mL biotin met the acceptance criteria of ≤10%, indicating that the assay was not susceptible to excess levels of biotin.

### 3.4. Influence of Age on NT-proBNP Levels

Median NT-proBNP of subjects < 50 years old was 43.0 pg/mL (range 4.9–456 pg/mL); for those 50–75 years old it was 95.1 pg/mL (range 10.5–1079 pg/mL), and for those >75 years old it was 173.1 pg/mL (range 23.2–1948 pg/mL) (see Figure 3). There was a significant difference between the median NT-proBNPs of each age group (<50 and 50–75: median difference 47.7 pg/mL, 95% CI 36.2–61.4, *p* < 0.0001; 50–75 vs. >75: median difference 68.5 pg/mL, 95% CI 35.5–102.1, *p* < 0.0001). The 99th percentile of each age group was 399/846/1858 pg/mL in the <50/50–75/>75 age groups.

## 4. Discussion

We have demonstrated that the Abbott Architect NT-proBNP assay shows good performance, in agreement with the manufacturer’s claims. Furthermore, this was confirmed over two sites in different geographical locations. Our evaluated precision is similar to those reported in another study [29] that evaluated the between-run precision of the six NT-proBNP assays (level 1 1.5–7.3%, level 2 1.6–6.9%, level 3 1.6–9.4%). In our study, we found that the Roche NT-proBNP had a slightly persistent negative bias compared to the Abbott NT-proBNP, with the bias percentage decreasing with increasing NT-proBNP level. At NT-proBNP levels of <125 pg/mL, the bias was 25%, while at >125 pg/mL, it was <15%. This is fairly similar to some studies comparing other NT-proBNP assays: for the ADVIA NT-proBNP, the average positive bias was 17.8%, and at 125/300 ng/mL, the bias was 22.5%/26.3% [30]. While not fully interchangeable, the Abbott assay shows excellent agreement with the Roche assay (r = 0.999), even beyond Roche NT-proBNP levels of 30,000 pg/mL.

Both the ESC [5], AHA [31], and the ICON study [24] recommend age-specific cut-offs for NT-proBNP to rule-in acute heart failure: namely, 450 pg/mL for <50 years, 900 pg/mL for 50–75 years, and 1800 pg/mL for >75 years. This has been confirmed in real-world studies demonstrating that different NT-proBNP “rule-in” cut-off points are required to optimise the diagnosis of HF in different age groups, such as in the ICON RELOADED study [32]. We confirm that, using the same age groups as the ICON study, there were significant differences in NT-proBNP levels between the age groups with the Abbott NT-proBNP assay as well. The 99th percentile NT-proBNP in each group is close to the Roche cut-offs used in the ICON study (399/846/1858 pg/mL for <50 y/50–75 y/>75 y). Furthermore, the bias between the Roche and Abbott assays at the aforementioned Roche cut-offs (450/900/1800 pg/mL) was also quite acceptable (<20%) in our study (see Table 1). Our findings suggest that similar age group cut-offs can be used to rule-in acute heart failure on the Abbott assay. However, it is important to remember that other assays may have a larger bias when compared to either the Roche/Abbott assays [33], and as such, the suitability of the ICON age-specific cut-offs should be evaluated for each assay accordingly. In addition, there remain ranges where the significance of a NT-proBNP value is uncertain, for example, between 300–1800 pg/mL in patients over 75 years old [34]. Thus, even with age-specific cut-offs, NT-proBNP results must still be interpreted in the context of the clinical picture, and with other supporting investigations.

As a non-competitive (sandwich) immunoassay, biotin can interfere in the formation of reactant complexes in these assays, resulting in falsely low results [35]. We confirm that biotin up to 4250 ng/mL does not affect the Abbott assay. This biotin threshold is also reported by the IFCC [23]; they also noted that other NT-proBNP assays have lower thresholds for susceptibility to the biotin effect (30–1500 ng/mL). The Abbott assay is, thus, the least likely to be affected by high biotin concentrations. Roche claims that their NT-proBNP is unaffected by biotin levels up to 1250 ng/mL. However, some studies [36] have shown that the Roche NT-proBNP assay was not significantly affected by serum biotin levels of up to 3600 pg/mL, with only a slight decrease of 3.7 pg/mL in the NTproBNP level. Further studies may be required to verify the limit of biotin interference on the Roche NT-proBNP assay. However, such high levels of biotin are uncommon [37], and it would be of greater concern only in patients taking large doses of biotin for the treatment of multiple sclerosis and/or if patients have severe end-stage renal disease due to the impaired clearance of biotin. Furthermore, it must be remembered that even for the same dose of biotin, inter-individual serum biotin levels are highly variable, and it is difficult to predict the precise impact of biotin interference [38]. Different laboratories have proposed varying methods for biotin depletion to overcome this interference, including the use of pooled streptavidin-coated magnetic microparticles [39], with good levels of recovery for NT-proBNP (95% confidence interval for recovery 91.2–95.9%).

We report the following novel findings in our study:We confirm the assay’s good performance across two sites in two different geographical locations;We report the persistent negative bias between the Roche and Abbott assays (Roche < Abbott) across different levels of NT-proBNP;The AHA/ESC/ICON age-related NT-proBNP cut-offs to rule-in acute heart failure is also applicable for use on the Abbott NT-proBNP assay.

One limitation of our study is that we were unable to compare the Abbott Architect NT-proBNP assay to other NT-proBNP assays, including the newer NT-proBNP point-of-care tests. However, other studies [29] have shown a good agreement between other NT-proBNP point-of-care tests with the Roche Elecsys NT-proBNP assay (Roche Cobas H232 point-of-care assay; Pearson correlation coefficient 0.983), which, in turn, has an excellent agreement with the Abbott assay, as demonstrated. As we only had access to de-identified leftover patient sera, we were unable to determine how being overweight/obese and having type 2 diabetes mellitus was operative in our study participants, nor could we determine their impact on the Abbott NT-proBNP assay. This is certainly a worthwhile endeavour for a future study. However, extensive data is available on this issue for the Roche NT-proBNP assay, and we expect the Abbott assay to perform similarly in these patient groups, given the close agreement between these two assays. We only studied a small cohort of cardio-renal healthy subjects to assess the applicability of NT-proBNP age cut-offs for the diagnosis of acute heart failure, and we did not have access to samples from patients having acute decompensated HF. Further larger studies using sera from acute heart failure patients would be desirable to confirm diagnostic age-specific cut-offs to rule-in acute heart failure.

## 5. Conclusions

In conclusion, we report that the Abbott Architect NT-proBNP assay performs well and within the manufacturer’s claims for assay precision, linear range, limit of blank, limit of quantitation, and biotin interference levels. This is further strengthened by the performance evaluations being performed across two centres in different geographic locations. Our findings also lend support to age-stratified reference and risk levels for the Abbott NT-proBNP using ICON/ESC/AHA age groups, and the varying bias between assays at different levels of NT-proBNP.

## Figures and Tables

**Figure 1 diagnostics-12-01172-f001:**
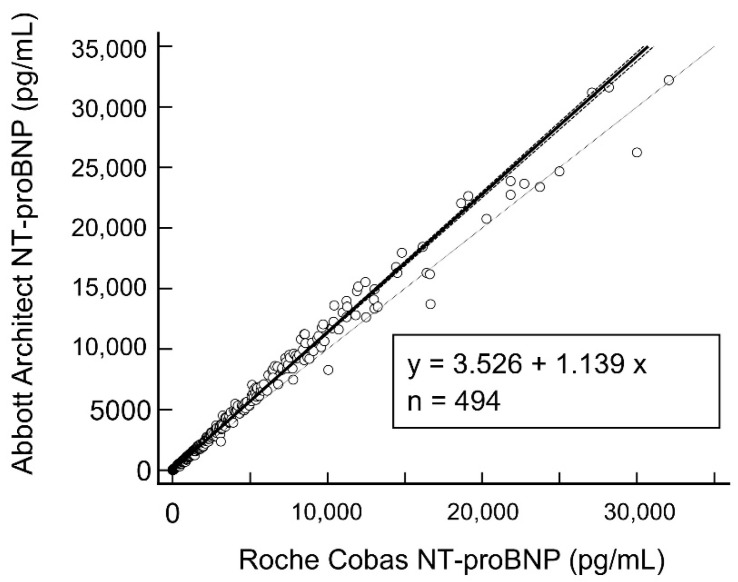
Passing–Bablok regression between the Abbott Architect and Roche Cobas NT-proBNP assays. Abbreviations: NT-proBNP—N-terminal pro-brain natriuretic peptide.

**Figure 2 diagnostics-12-01172-f002:**
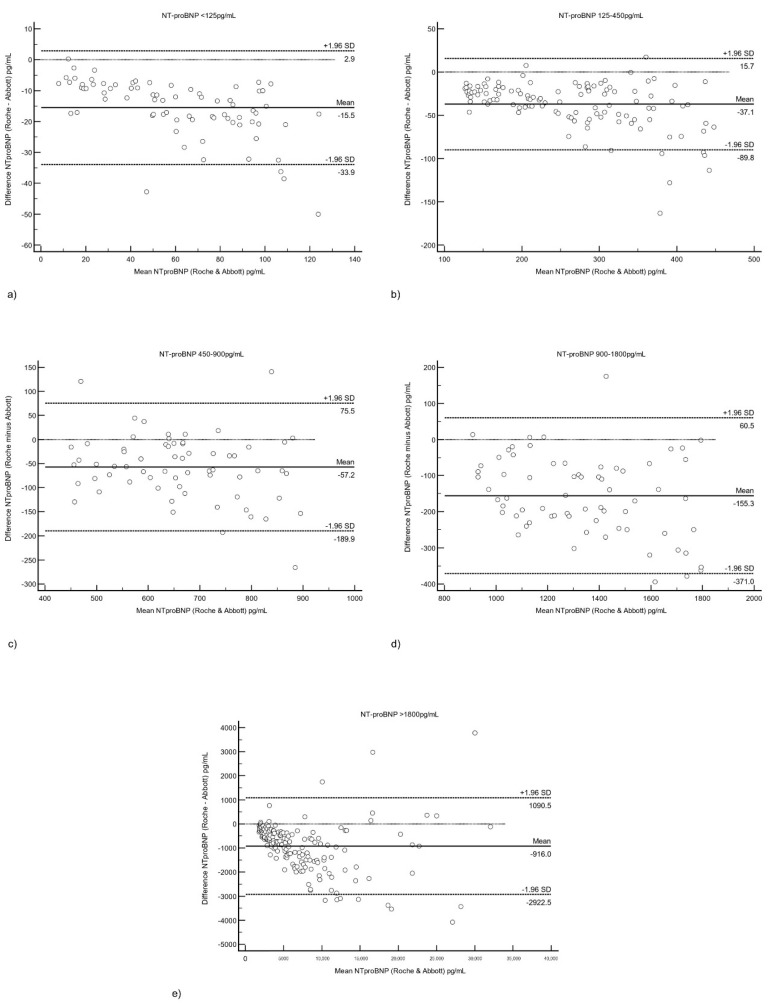
Bland–Altman analysis of the bias between the Roche and Abbott NT-proBNP assays at diagnostic cut-offs of Roche NT-proBNP. (**a**) <125 pg/mL and (**b**) 125–450 pg/mL, and according to ESC/AHA age-related cut-offs of (**c**) 450–900 pg/mL, (**d**) 900–1800 pg/mL, (**e**) >1800 pg/mL. Abbreviations: NT-proBNP: N-terminal pro-brain natriuretic peptide.

**Figure 3 diagnostics-12-01172-f003:**
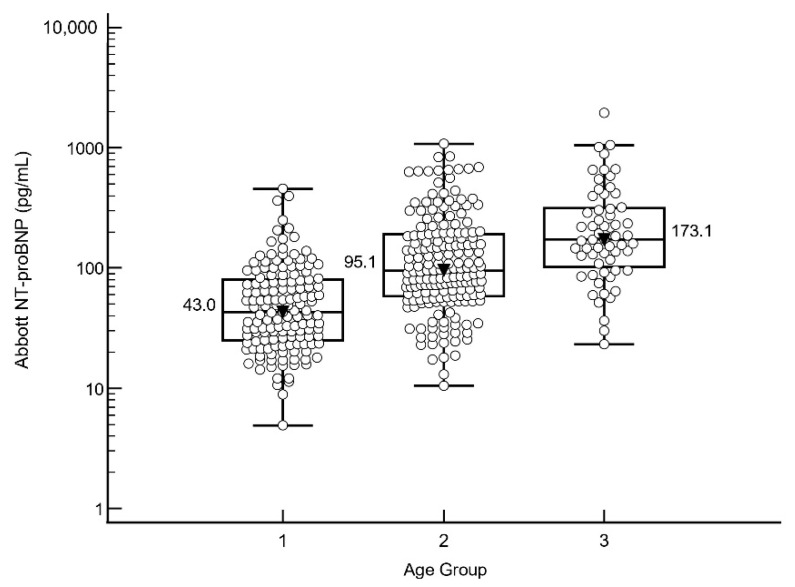
Abbott Architect NT-proBNP levels according to age groups. Age group 1: <50 years old, 2: 50–75 years old, and 3: >75 years old. Median NT-proBNP levels for each group is displayed. Abbreviations: NT-proBNP: N-terminal pro-brain natriuretic peptide.

**Table 1 diagnostics-12-01172-t001:** Bias as a percentage of the mean NT-proBNP at different levels of Roche NT-proBNP.

Roche NT-proBNP Level (pg/mL)	N	Mean NT-proBNP (pg/mL)	Bias (pg/mL), (% of Mean NT-proBNP)
<125	75	60.9	15.5 (25.5%)
125–450	116	263	37.1 (14.1%)
450–900	65	663	57.2 (8.63%)
900–1800	68	1338	155 (11.6%)
>1800	170	7380	916 (12.4%)

Abbreviations: NT-proBNP: N-terminal pro-brain natriuretic peptide.

**Table 2 diagnostics-12-01172-t002:** Biotin recovery on the Abbott Alere NT-proBNP assay for the Architect analyser with increasing levels of biotin.

Sample	Biotin (ng/mL)	Average Result (pg/mL)	Average Result (pmol/L)	% Difference vs. Un-Spiked (0) Biotin Sample
**1**	0	79.30	9.36	
	531.3	79.43	9.37	0.17%
	1062.5	78.44	9.26	−1.09%
	1593.8	79.89	9.43	0.74%
	2125.0	78.54	9027	−0.96%
	2656.3	80.24	9.47	1.18%
	3187.5	78.79	9.30	−0.65%
	3718.8	78.80	9.30	−0.64%
	4250.0	79.74	9.41	0.55%
**2**	0	1815.26	214.20	
	531.3	1760.91	207.79	−2.99%
	1062.5	1759.66	207.64	−3.06%
	1593.8	1777.98	209.80	−2.05%
	2125.0	1797.01	212.05	−1.01%
	2656.3	1804.90	212.98	−0.57%
	3187.5	1783.38	210.44	−1.76%
	3718.8	1765.07	208.28	−2.76%
	4250.0	1830.18	215.96	0.82%

## Data Availability

The datasets generated during and/or analysed during the current study are not publicly available due to privacy issues and national laws but are available from the corresponding author on reasonable request under the provision that data may not leave the hospital/center premises.

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
