# Peer review of "Performance of the Abbott Architect Immuno-Chemiluminometric NT-proBNP Assay"

_diagnostics, 2022, doi:10.3390/diagnostics12051172_

Round 1
Reviewer 1 Report
The authors submitted a research article with the aim of evaluating the performance of the Abbott NT-proBNP assay against the Roche NT-proBNP immunoassay across two sites. They compared 494 samples and found good agreement between the Roche and Abbott NT-proBNP assays. The aim of the study and has scientifically important sound. I found the hypothesis original, which deserved being investigated. The topic is relevant in the field of “Clinical Diagnostics”. The findings added novelty to the recent data affecting the use of kits manufactured by Abbott and Roche in HF patients with diagnostic purpose, but not for point-of-care therapy. The readers may find resoundingly clear scientific proof of well agreement of the results of NT-proBNP measures received with Abbott NT-proBNP assay. The manuscript has legible and clear tables and figures. Along with it, its structure is logically clear and the paper is well referenced. The conclusion corresponds to the results and discussion and rises up the issues that are attractive for readers. I have no serious flaws to hypothesis, structure of manuscript, statistics as well as results and discussion. Although the data appear to be clinically important, I would like to put forward several questions to discuss.
- The authors should add a short commentary about keeping the kit in fit to point-of-care management of HF along with clear clinical recommendation for use.
- The authors reported a clear agreement of the results of NT-proBNP measurements in subgroups patients with different ages, but not those who had overweigth / obesity, T2DM, CKD. Please, discuss it as a bias in the section Study limitations.
Conclusion: this article deserves being accepted after minor revision.
Author Response
Response to Review 1
- The authors should add a short commentary about keeping the kit in fit to point-of-care management of HF along with clear clinical recommendation for use.
We have alluded to the expected performance of the Abbott NTproBNP assay in the last paragraph of the discussion section on page 9 (marked in yellow).
- The authors reported a clear agreement of the results of NT-proBNP measurements in subgroups patients with different ages, but not those who had overweigth / obesity, T2DM, CKD. Please, discuss it as a bias in the section Study limitations.
As we only had access to de-identified leftover patient sera we were unable to determine how overweight/obesity and type 2 diabetes mellitus was operative in our study participants and their impact on the Abbott NT-proBNP assay. We did not intend to study the influence of CKD on the Abbott NT-proBNP levels. These studies are certainly worthwhile for a future study. However, extensive data is available on these issues for the Roche NT-proBNP assay and we expect the Abbott assay to perform similarly in these patient groups given the close agreement between these two assays. (See section of discussion marked in blue).
Reviewer 2 Report
The Authors evaluated the performance of the Abbott Architect immune-chemiluminometric NT-proBNP assay in comparison with that of Roche diagnostics. The method is appropriate and the results clearly presented. Some points could be considered:
- the authors enrolled apparently healthy subjects. However, the level of NT-proBNP detected are very high in some cases. This makes very likely the presence of patients affected by cardiovascular diseases.
More details about the selection of samples should be provided. In particular, it should be reported if data about patients' symptoms were evaluated. This is even more relevant considering that the evaluation of NT-proBNP should be performed only when heart failure is suspected.
- NT-proBNP is also used in order to monitor heart failure hemodynamic stability. Do the results of the study provide some information about this topic?
Other specific comments-
The main original aspect of the paper is related to the investigation of any age- related cut-offs for the Abbott NT-proBNP assay in comparison with Roche Diagnostics. The results could be useful to the application of current guidelines recommendations when the Abbott system is used.
The specific gap addressed by the paper is of interest mainly for clinical pathologists, whereas the interest for cardiologists is very limited.
The additive information is related to the analysis according to the age of patients.
-More details about the subjects enrolled to test serum levels of NT-ptoBNP should be provided in order to better clarify clinical usefulness of the results.
Author Response
Response to Reviewer 2
- The authors enrolled apparently healthy subjects. However, the level of NT-proBNP detected are very high in some cases. This makes very likely the presence of patients affected by cardiovascular diseases. More details about the selection of samples should be provided. In particular, it should be reported if data about patients' symptoms were evaluated. This is even more relevant considering that the evaluation of NT-proBNP should be performed only when heart failure is suspected.
The main study was to evaluate the analytical performance of the new Abbott NT-proBNP compared to the established Roche assay. The 388 “healthy” subjects did not have a history of heart disease and NTproBNP was not requested by their attending doctors; thus presumably asymptomatic. We used deidentified leftover sera for these studies and they had healthy renal function (eGFR>90mL/min); more clinical details are not available to us.
- NT-proBNP is also used in order to monitor heart failure hemodynamic stability. Do the results of the study provide some information about this topic?
Other than random cases with high NTproBNP used for correlation studies between the two assays, we did not study any subjects with heart failure. This limitation was highlighted in the last two sentences of the final paragraph of the discussion section.
- The specific gap addressed by the paper is of interest mainly for clinical pathologists, whereas the interest for cardiologists is very limited.
The additive information is related to the analysis according to the age of patients.
This was our original intent – to assess the analytical performance of the new Abbott NTproBNP assay. While clinical pathologists will naturally be interested in such technical matters we hope that cardiologists will also be interested when they perform studies, when their central labs change assays, and when they see patients with NTproBNP results done on the Abbott assay.
- More details about the subjects enrolled to test serum levels of NT-ptoBNP should be provided in order to better clarify clinical usefulness of the results.
See answer in comment 1.